# Regulation of X-linked gene expression during early mouse development by *Rlim*

Feng Wang[1†], JongDae Shin[1,2†], Jeremy M Shea[3], Jun Yu[1], Ana Bošković[3], Meg Byron[4], Xiaochun Zhu[1], Alex K Shalek[5,10,6,7], Aviv Regev[6,11,8], Jeanne B Lawrence[4], Eduardo M Torres[1], Lihua J Zhu[1,9], Oliver J Rando[3], Ingolf Bach[1]*

[1]Department of Molecular, Cell and Cancer Biology, University of Massachusetts Medical School, Worcester, United States; [2]Department of Cell Biology, College of Medicine, Konyang University, Daejeon, Korea; [3]Department of Biochemistry and Molecular Pharmacology, University of Massachusetts Medical School, Worcester, United States; [4]Department of Cell and Developmental Biology, University of Massachusetts Medical School, Worcester, United States; [5]Department of Chemistry, Massachusetts Institute of Technology, Cambridge, United States; [6]Broad Institute of MIT and Harvard, Cambridge, United States; [7]Ragon Institute of MGH, MIT and Harvard, Cambridge, United States; [8]Howard Hughes Medical Institute, Massachusetts Institute of Technology, Cambridge, United States; [9]Program in Bioinformatics and Integrative Biology, University of Massachusetts Medical School, Worcester, United States; [10]Institute for Medical Engineering and Science, Massachusetts Institute of Technology, Cambridge, United States; [11]Department of Biology, Massachusetts Institute of Technology , Cambridge, United States

*For correspondence: ingolf.
bach@umassmed.edu

†These authors contributed
equally to this work

Competing interest: See
page 15

Reviewing editor: Kevin Struhl,
Harvard Medical School, United
States

**Abstract** Mammalian X-linked gene expression is highly regulated as female cells contain two and male one X chromosome (X). To adjust the X gene dosage between genders, female mouse preimplantation embryos undergo an imprinted form of X chromosome inactivation (iXCI) that requires both *Rlim* (also known as Rnf12) and the long non-coding RNA *Xist*. Moreover, it is thought that gene expression from the single active X is upregulated to correct for bi-allelic autosomal (A) gene expression. We have combined mouse genetics with RNA-seq on single mouse embryos to investigate functions of *Rlim* on the temporal regulation of iXCI and *Xist*. Our results reveal crucial roles of *Rlim* for the maintenance of high *Xist* RNA levels, *Xist* clouds and X-silencing in female embryos at blastocyst stages, while initial *Xist* expression appears *Rlim*-independent. We find further that X/A upregulation is initiated in early male and female preimplantation embryos.

## Introduction

Most mammalian cells contain two transcriptionally active copies of autosomal chromosomes but only one active X chromosome (X). This is due to the fact that female cells inactivate one X in a process known as X chromosome inactivation (XCI), to correct for male/female (F/M) gene dosage imbalances caused by the presence of two X chromosomes. In addition, to adjust for X/autosomal (X/A) imbalances arising from transcription of both autosomal copies of most genes, it is thought that male and female cells upregulate gene expression from the single active X two-fold.

Beginning at the 4-cell stage in female mouse embryos, imprinted XCI (iXCI) exclusively silences the paternally inherited X (Xp), and this pattern of XCI is maintained in extraembryonic trophoblast

cells. In contrast, epiblast cells in the inner cell mass (ICM) of blastocysts that will give rise to the embryo reactivate the Xp and undergo a random form of XCI (rXCI) around implantation (E5-E5.5) (*Disteche, 2012*; *Payer and Lee, 2014*; *Galupa and Heard, 2015*). During XCI the long non-coding RNA *Xist* progressively paints the X chromosome from which it is synthesized, thereby triggering repressive histone modifications including H3K27me3 (*Plath et al., 2003*) and transcriptional silencing of X-linked genes (*Disteche, 2012*; *Payer and Lee, 2014*; *Galupa and Heard, 2015*). *Xist* is required both for iXCI and rXCI (*Marahrens et al., 1997*; *Penny et al., 1996*). The X-linked gene *Rlim* encodes a ubiquitin ligase (*Ostendorff et al., 2002*) that shuttles between the nucleus and cytoplasm (*Jiao et al., 2013*) and modulates transcription via regulating nuclear multiprotein complexes (*Bach et al., 1999*; *Güngör et al., 2007*). *Rlim* promotes the formation of *Xist* clouds both during iXCI in female mice (*Shin et al., 2010*) and in female embryonic stem cells (ESCs) undergoing rXCI in culture (*van Bemmel et al., 2016*). In mice, however, *Rlim* is dispensable for rXCI in epiblast cells (*Shin et al., 2014*). While both *Rlim* and *Xist* play essential roles during iXCI in mice (*Marahrens et al., 1997*; *Shin et al., 2010*), questions remain with regards to their functions on the general kinetics of X-linked gene expression, their functional interconnection and the contribution of maternally vs embryonically expressed RLIM for iXCI. Moreover, while evidence for X/A upregulation was observed not only in adult mouse tissues but also in mouse ES cells and epiblast cells at blastocyst stages (*Deng et al., 2013*, *2011*, *2007*), details on the developmental X upregulation are lacking.

We have carried out RNA-seq on single mouse embryos to investigate the regulation of X-linked gene expression during pre- and peri-implantation development by *Rlim*. Analyses of WT, *Rlim*-knockout (KO) or *Xist*-KO embryos reveal that *Rlim* is required for maintenance of *Xist* clouds and iXCI in female embryos at blastocyst stages. In addition, our data uncover that X dosage compensation via iXCI in female mice occurs concurrently with a general X/A upregulation in both male and female preimplantation embryos. These results represent a comprehensive view on the regulation of X-linked gene expression during early mouse embryogenesis by *Rlim* and *Xist*.

## Results

### Elucidation of the mouse preimplantation transcriptome by single embryo RNA-seq

Several studies have investigated the dynamics of XCI and X-silencing in female ES cells (*Lin et al., 2007*; *Marks et al., 2015*) or trophoblasts (*Calabrese et al., 2012*) using RNA-seq. To assess the general dynamics of X-linked gene expression and iXCI in vivo, we have adapted single cell RNA-seq technology (*Shalek et al., 2013*; *Jaitin et al., 2014*) to elucidate the pre- and peri-implantation transcriptome using single mouse embryos (*Sharma et al., 2016*). This is a valid strategy because (1) early embryos consist of a limited number of cell types: essentially one totipotent cell type up to E3, and three cell types at blastocyst stages - epiblast, trophoblast and primitive endoderm cells - that express known marker genes (2) cells of preimplantation embryos undergo iXCI that exclusively silences the Xp and (3) in mice there is only a relatively small number of genes that escape XCI (*Yang et al., 2010*; *Finn et al., 2014*). Thus, using RNA-seq we examined embryos at the 4- and 8-cell stages, early and late morula (E2.5 and E3.0), and blastocyst stages (E3.5, E4.0 and E4.5), comparing global changes in gene expression. We also included trophoblasts isolated from blastocyst outgrowths of cultured E4.0 embryos in our analyses. In contrast to previous studies on single cells of preimplantation embryos that were performed in a mixed genetic background (*Deng et al., 2014*), embryos were generated in a C57BL/6 background to exclude potential background influences on the general kinetics of iXCI and/or X upregulation. Moreover, because the known functions of *Rlim* during mouse embryogenesis are restricted to XCI in females (*Shin et al., 2010*, *2014*), RNA-seq experiments were performed on WT and *Rlim*KO embryos. Based on a mouse model allowing a conditional KO (cKO) of the *Rlim* gene, we have previously generated females carrying a maternally transmitted *Sox2*-Cre-mediated cKO allele and a paternally transmitted germline KO allele (*Rlim*$^{cKOm/\Delta p-SC}$). These females appear healthy, fertile and lack RLIM in all somatic tissues as well as their germline (*Figure 1A,B*) (*Shin et al., 2014*). Thus, to efficiently generate male and female germline *Rlim* KO embryos (Δ/Y, Δ/Δ) and to eliminate potential influences of maternal RLIM in oocytes on the iXCI process, all *Rlim* KO embryos were generated by crossing *Rlim*$^{cKOm/\Delta p-SC}$ females with

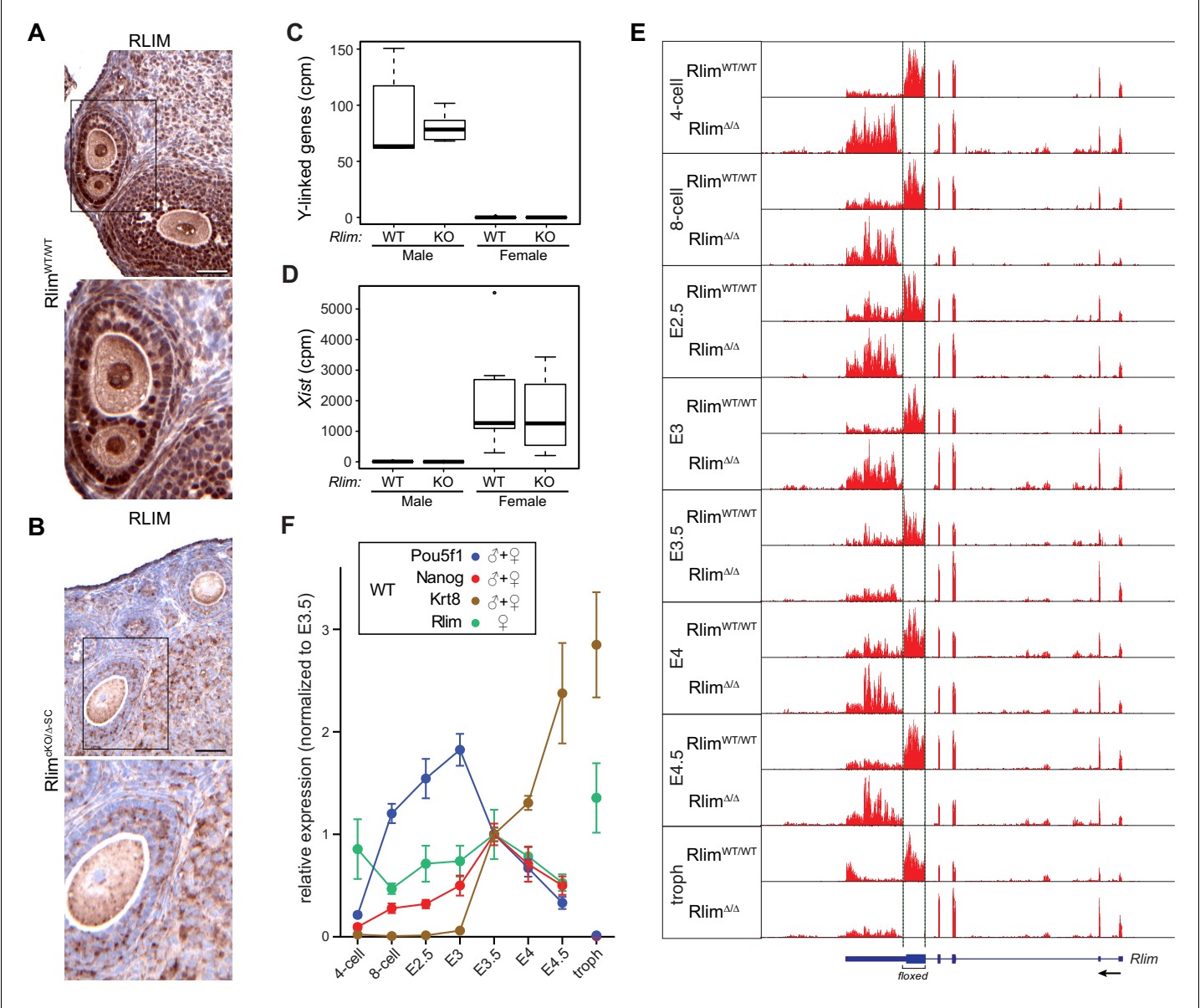

**Figure 1.** Elucidating the transcriptome of mouse pre/peri-implantation development via RNA-seq of single embryos WT and KO for *Rlim*. All *Rlim* germline KO (Δ) embryos were generated by *Rlim^cKO/Δ-SC* x *Rlim^Δ/Y* crosses. Embryonic stages are indicated, troph = trophoblasts. (**A, B**) Lack of RLIM in oocytes of *Rlim^cKO/Δ-SC* females. Immunohistochemical stainings of representative ovarian sections of adult *Rlim^WT/WT* (**A**) and *Rlim^cKO/Δ-SC* (**B**) females (n = 3, each) using antibodies directed against RLIM. Scale bars, 60 μm. Boxed regions are shown in higher magnification below. Note the lack of RLIM immunoreactivity in nuclei and pronuclei of both somatic cell types and oocytes in *Rlim^cKO/Δ-SC* females, respectively. (**C, D**) Gender determination of embryos in RNA-seq on whole preimplantation embryos WT and KO for *Rlim*. As example, the distributions of reads at the 8-cell stage normalized to autosomes of Y-linked genes (**C**) and *Xist* (**D**) are shown in box-plots. Note that embryos with high levels of Y-linked genes display low levels of *Xist* and were therefore categorized as males, whereas those with low and high levels of Y-linked genes and *Xist*, respectively, were categorized as females. cpm = counts per million mapped reads. (**E**) Modified from the UCSC Genome Browser: Cumulative mapped raw reads on the *Rlim* locus of pooled embryos WT/WT or Δ/Δ for *Rlim* (females only) at all developmental stages (variable scales). Structure of the *Rlim* gene is shown below in blue with boxed exon regions. Protein coding regions are indicated in thicker stroke. Arrow indicates direction of transcription. Floxed area deleted in the *Rlim* cKO is indicated. Note the lack of reads in the floxed area of *Rlim^Δ/Δ* females. This was also true for male *Rlim^Δ/Y* embryos (data not shown). (**F**) Developmental profile of relative expression of selected single genes in WT embryos. Data representing *Oct4/Pou5f1* and *Nanog* (ES cell markers) and *Krt8* (trophoblast) were pooled from WT females and males. *Rlim* data were collected from WT/WT females only. Reads were normalized to those at stage E3.5, because all of the selected genes are expected to be active at this stage.

The following figure supplement is available for figure 1:

*Figure 1 continued on next page*

*Figure 1 continued*

**Figure supplement 1.** Details of RNA-seq experiments on single embryos WT and KO for *Rlim*.

germline KO males ($Rlim^{\Delta/Y}$) (*Figure 1—figure supplement 1A*). RNA was prepared and barcoded RNA-seq libraries were constructed from 187 samples that were distributed between two 96-well plates, each with similar numbers of embryos from each stage and mating. Libraries were pooled and sequenced to an average read depth of 2.95 million reads. This read depth lies within the range typically obtained by using single-cell technology (*Shalek et al., 2013*). The gender of each embryo was determined by assessing expression of Y-linked genes, which occurs only in male embryos, and *Xist*, which is expressed only in females (*Figure 1C,D*). 12 samples for which gender could not be assigned or displayed less than 280,000 total reads were removed, leaving 175 samples for further analysis (*Figure 1—figure supplement 1B*; *Supplementary file 1*). Mapping reads from WT and KO embryos to the *Rlim* locus showed that the deleted region in KO embryos was not represented (*Figure 1E*; data not shown), validating both the mating strategy and the specificity of the data obtained via whole embryo RNA-seq. Because of expected differences in expression of X-linked genes during preimplantation development between genders and between females WT and KO for *Rlim*, each library was normalized to its total autosomal gene expression rather than expression of all genes on chromosomes. As expected, comparing Log2-transformed data of 10552 annotated genes that are expressed at all examined developmental stages revealed no significant differences in gene expression at the chromosome level between $Rlim^{\Delta/Y}$ and WT/Y males (*Figure 1—figure supplement 1C*), consistent with the finding that male mice lacking *Rlim* develop normally (*Shin et al., 2010*). Analyses of WT embryos (pooled male and female embryos for each stage), showed that the developmental transcript profiles matched well with expression of epiblast markers *Pou5f1* (Oct4) (*Rosner et al., 1990*; *Schöler et al., 1990*) and *Nanog* (*Chambers et al., 2003*; *Mitsui et al., 2003*) and trophoblast marker *Krt8* (Troma1) (*Brûlet and Jacob, 1982*) (*Figure 1F*). The observed variations between replicates were well within margins previously observed for single cell RNA-seq technology (*Jaitin et al., 2014*; *Shalek et al., 2013*), and comparison of expression levels of these cell markers between $Rlim^{\Delta/\Delta}$ and WT/WT females suggested that the general cell type specification events that occur at blastocyst stages were similar in embryos of both genotypes (*Figure 1—figure supplement 1D*). Because *Rlim* is X-linked and its functions appear restricted to females (*Shin et al., 2010*; *Jiao et al., 2012*), expression of this gene was evaluated in female WT/WT embryos only. *Rlim* RNA levels decreased from the 4-cell to the 8-cell stage, possibly due to depletion of maternal pools, then gradually increased up to E3.5, and decreased thereafter (*Figure 1F*). High relative levels of *Rlim* were detected in trophoblasts, suggesting that the cells of the ICM are mostly responsible for the diminished *Rlim* levels in E4.5 embryos, consistent with a drop in RLIM protein levels specifically in epiblast cells (*Shin et al., 2014*). Random subsampling of libraries to 200,000 reads per embryo (*Robinson and Storey, 2014*) showed similar results (*Figure 1—figure supplement 1E*), indicating that the sequencing depth was not a limiting factor in our data analyses.

### *Rlim* is required for upregulation of *Xist* and maintenance of *Xist* clouds in female embryos

*Xist* RNA levels were very low in males at all stages (*Figure 2A*), as expected. Much higher levels of *Xist* were measured in females, although variations among replicates at each stage were high relative to those obtained for *Pou5f1 (Oct4)*, *Nanog* and *Krt8*, likely reflecting biological variations in individual animals in addition to the technical variability of single-embryo RNA-seq (compare *Figures 1F* and *2A*). Because the *Xist* gene is located within the *Tsix* gene, which is transcribed in the antisense orientation (*Lee and Lu, 1999*), we examined the locations of reads mapping to this locus. We detected a high read density located precisely within the *Xist* transcription unit (>90% of reads) and very few reads in introns or exons specific to *Tsix* (*Figure 2—figure supplement 1A*), indicating that the detected reads mostly correspond to *Xist* RNA. *Xist* levels in WT female embryos increased dramatically from the 4-cell to the 8-cell stages and peaked around E3.5 (*Figure 2A*). Consistent with the detection of *Xist* transcription foci in early *Rlim* KO female embryos (*Shin et al., 2010*), the developmental onset of *Xist* was similar in $Rlim^{\Delta/\Delta}$ females, but significantly lower levels

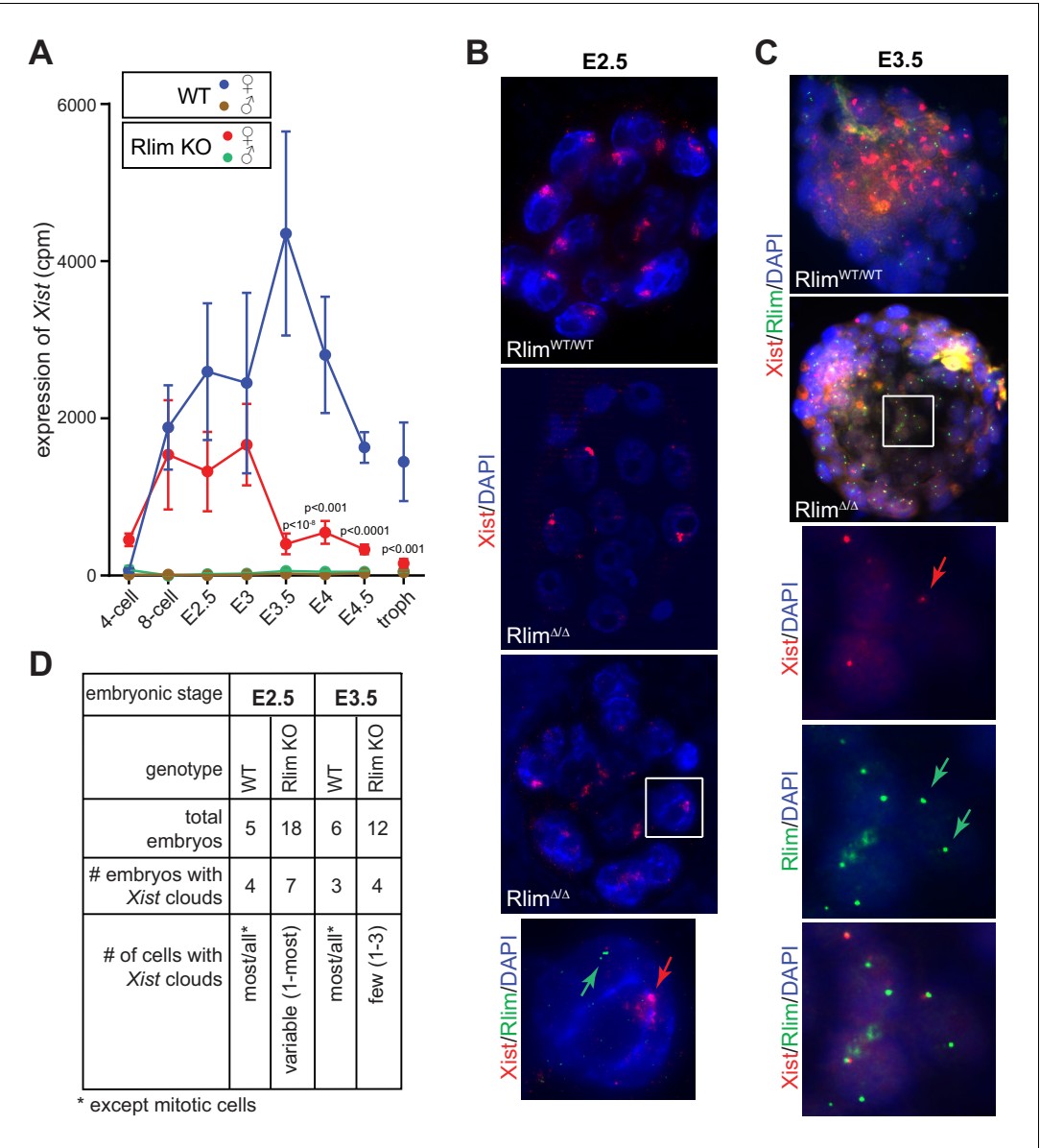

**Figure 2.** *Rlim* is required for the maintenance of *Xist* expression and *Xist* clouds at blastocyst stages. (**A**) Expression profiles from the *Xist* locus in preimplantation embryos WT and KO for *Rlim*. Error bars indicate standard error of the mean (SEM). Significant differences of *Xist* levels in WT and KO females P<0.01 are indicated (Student's t-test). (**B**) WT and *Rlim*KO embryos at E2.5 were co-stained with probes recognizing *Xist* (red) and *Rlim* (green) via RNA FISH. Two representative *Rlim*[Δ/Δ] embryos are shown. The boxed area is magnified in the panel below. Arrows point at *Xist* cloud and *Rlim* transcription focus. (**C**) RNA FISH on WT and *Rlim*KO embryos at E3.5. The boxed area is magnified in the panels below. Arrows point at *Xist* and *Rlim* transcription foci. (**D**) Tabular summary of *Xist* clouds detected in **B** and **C**.

The following figure supplement is available for figure 2:

**Figure supplement 1.** *Rlim* regulates *Xist* levels and *Xist* clouds at blastocyst stages.

of *Xist* were measured only after E3.5 (*Figure 2A*). Random subsampling of libraries to 200,000 reads per embryo (*Robinson and Storey, 2014*) showed a similar *Xist* expression profile (*Figure 2— figure supplement 1B*). Moreover, strand-specific RT-qPCR confirmed that *Rlim*[Δ/Δ] females expressed similar levels of *Xist* when compared to WT/WT at E2.5, but much lower levels at E3.5

and thereafter (*Figure 2—figure supplement 1C*). These results prompted us to examine *Xist* clouds in females at E2.5 morula and E3.5 blastocyst stages, comparing *Rlim*$^{\Delta/\Delta}$ with WT/WT embryos. This was done via RNA FISH, co-staining with probes against *Rlim,* which recognizes both the WT and KO transcripts (*Shin et al., 2010*), and *Xist*. As expected, *Xist* clouds were detected in WT embryos of both embryonic stages (*Figure 2B–D*). In these female embryos most/all cells exhibited clouds except those undergoing mitosis. *Xist* clouds were also detected in a significant number of *Rlim* KO embryos at both developmental stages. In contrast to WT/WT, however, in *Xist* cloud-positive *Rlim*$^{\Delta/\Delta}$ embryos at E2.5, the number of cells displaying clouds was highly variable, ranging between 1 cell to most cells (*Figure 2B,D*). As few cells of 8-cell staged female embryos lacking RLIM display *Xist* clouds (*Shin et al., 2010*), these results indicate that *Rlim*$^{\Delta/\Delta}$ females develop *Xist* clouds at morula stages, but likely with a slower kinetics when compared to WT/WT females. As expected, at E3.5 the vast majority of cells exhibited transcription foci for *Xist* and only few cells displayed *Xist* clouds in *Rlim*$^{\Delta/\Delta}$ embryos (*Figure 2C,D*). Combined, these data reveal that *Rlim* is dispensable for initial *Xist* expression in 4-cell stage embryos but required for upregulation/maintenance of *Xist* expression after E3. They further show that *Xist* clouds transiently form in *Rlim*$^{\Delta/\Delta}$ female morulae, which cannot be maintained at blastocyst stages.

## *Rlim* regulates X-silencing during blastocyst stages

To examine the function of *Rlim* on global X-silencing during iXCI we compared normalized reads (FPKM) of all annotated X-linked genes from WT/WT or *Rlim*$^{\Delta/\Delta}$ female embryos with all male embryos generated in this dataset. In female WT/WT embryos the gene dosage was close to two-fold that of males at early developmental stages (likely due to two active X chromosomes) and then, starting at morula stages, gradually decreased to a gene dosage similar to that of males by late blastocyst stages, consistent with Xp silencing (*Figure 3A*). These results are in agreement with recent data obtained from allele-specific RNA-seq on single cells both from early mouse preimplantation embryos with a mixed genetic background (*Deng et al., 2014*; *Chen et al., 2016*) and human preimplantation embryos (*Petropoulos et al., 2016*). Consistent with the formation of transient *Xist* clouds (*Figure 2*) the X gene dosage in female *Rlim*$^{\Delta/\Delta}$ embryos was not significantly different from that of WT/WT females up to E3, revealing a partial silencing of X-linked genes at these early stages in the absence of RLIM. However, the continued further silencing of X-linked genes observed in WT/WT during blastocyst stages was significantly inhibited in *Rlim*$^{\Delta/\Delta}$ females (*Figure 3A*). Again, random subsampling of libraries for each embryo to 200.000 reads showed similar results (*Figure 3—figure supplement 1A*, and data not shown), demonstrating that these libraries are sequenced to a depth sufficient to accurately measure X-silencing. Considering an estimated average half-life of around 10 hr for mRNAs in mammalian cells (*Yang et al., 2003*) including mouse preimplantation embryos (*Kidder and Pedersen, 1982*), the detection of initial X-silencing at late morula/early blastocyst stages via RNA-seq on whole embryos (*Figure 3A*) fits well with published data obtained by RNA FISH that show diminished transcription foci of X-linked protein-encoding genes generally starting between the 8-cell and morula stages (*Patrat et al., 2009*; *Namekawa et al., 2010*). Moreover, the gradual X-silencing from the 8-cell stage to E4.5 in WT/WT females as determined by whole embryo RNA-seq (*Figure 3A*) is consistent with the existence of long-lived mRNAs in mouse preimplantation embryos (*Kidder and Pedersen, 1982*).

We next analyzed Log$_2$-transformed data of 10552 annotated genes that are expressed at all examined developmental stages. Comparing the developmental expression pattern between female *Rlim*$^{\Delta/\Delta}$ and male embryos revealed very few changes in autosomal gene expression due to the *Rlim* mutation, suggesting that the main function of *Rlim* in early embryonic gene regulation is in X-silencing (*Figures 3B*, *Figure 3—figure supplement 1B*). Indeed, examination of the silencing pattern of 351 annotated X-linked genes in WT females showed that gene silencing occurs within most regions on the X chromosome both during the early (8-cell to E3) and late phase (E3 to E4.5) of iXCI (*Figure 3C*). However, expression of several genes located in a region on the XqE3/F1 border is notably higher at early preimplantation stages in females WT and KO for *Rlim* when compared to males. Interestingly, this region overlaps a 1.1 Mb region that has been involved with meiotic regulation during spermatogenesis (*Zhou et al., 2013*). Moreover, levels of genes known to escape X-silencing during rXCI including *Kdm6a* and *Kdm5c* (*Berletch et al., 2010*) were also not significantly down-regulated during iXCI (not shown). As we are not able to distinguish mRNAs transcribed from the Xm or Xp, we cannot exclude the contribution of developmental transcriptional regulation

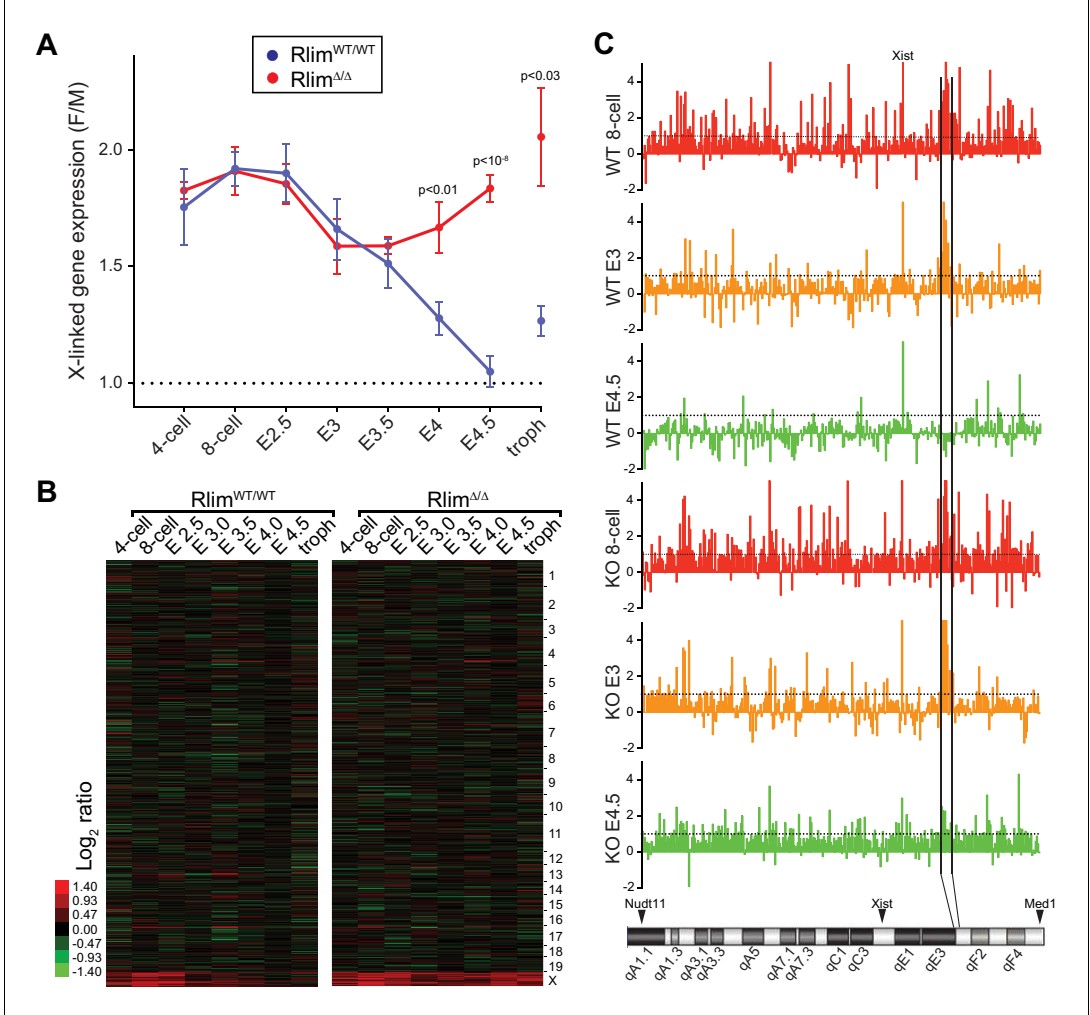

**Figure 3.** *Rlim* is required for X-silencing in females during blastocyst stages. Female expression data collected from *Rlim^{WT/WT}* or *Rlim^{Δ/Δ}* were compared with those of male (pooled KO and WT) embryos (F/M). Embryonic stages are indicated, troph = trophoblasts. (**A**) Developmental profile of X-silencing during iXCI in vivo as determined by comparing mean female/male (F/M) expression ratios of X-linked transcripts (minus *Xist*; in Fragments per kilobase of exon per million reads mapped (FPKM)). Error bars indicate SEM. Significant p values p<0.01 are indicated (Student's t-test). (**B**) Heat map representing Log$_2$ transformed data comparing global F/M mRNA expression level ratios from chromosomes (excluding the Y) of WT and KO embryos. Chromosomes corresponding to gene expression are indicated. (**C**) Gene silencing during iXCI occurs within most regions on the X chromosome. Log$_2$ F/M ratios of 351 X-linked genes at the 8-cell stage, E3 and E4.5 in WT and RlimKO are shown (values within 4.5 and -2). Horizontal dotted lines indicate Log$_2$ values of 1. The mouse X chromosome is shown below. Arrowheads indicate locations of the most centromeric (Nudt11) and most telomeric (Med1) genes included in this analysis. Expression and location of *Xist* is indicated. Expression of genes within a region indicated by vertical black lines is silenced late at blastocyst stages during iXCI.

The following figure supplements are available for figure 3:

**Figure supplement 1.** Rlim is required for X-silencing in female blastocysts.

**Figure supplement 2.** Comparison of X dosage compensation using F/M or allelic approach.

and/or effects of differential parental imprinting in these regions. To further examine X expression profiles in females, we compared the distributions of female E3/8-cell and E4.5/8-cell expression ratios of 755 X-linked genes that are expressed in at least 3 embryos. Results showed that there are little differences in expression profiles between E3/8-cell ratios of WT/WT and *Rlim^{Δ/Δ}* females (*Figure 3—figure supplement 1C*), whereas E4.5/8-cell ratios in *Rlim^{Δ/Δ}* females are generally two-fold higher when compared to WT/WT (*Figure 3—figure supplement 1D*). To verify global Xp silencing

we performed an independent RNA-seq experiment on 3 male and 3 female WT embryos with a heterozygous C57Bl/6 (B6) and castaneus/Eij (CAST) background (*Figure 3—figure supplement 2A*) allowing a direct comparison of data processed via F/M or by an allele-specific analysis. We chose embryos at E3.5 because at this stage about half of the X is silenced in female embryos (*Figure 3A*; *Figure 3—figure supplement 2B*). Indeed, in B6$_m$/CAST$_p$ embryos we measured only around 50% of X-linked CAST transcripts when compared to transcripts originating from the B6 X chromosome (*Figure 3—figure supplement 2C*). These results provide independent confirmation that the decline in F/M values of X-linked transcripts in females (*Figure 3A*) is due to X$_p$-silencing. These data corroborate our previous results and are consistent with (1) partial silencing of overall X-linked genes in female E3 embryos of both genotypes, and (2) the general silencing of one X chromosome in E4.5 WT/WT but not *Rlim*$^{\Delta/\Delta}$ females. Thus, our combined results distinguish an early *Rlim*-independent phase of iXCI in totipotent cells up to around E3 that leads to partial Xp-silencing, and a late phase after E3 that is *Rlim*-dependent and leads to robust Xp-silencing. These data further confirm that the process of iXCI progressively adjusts the X-linked gene dosage from the 8-cell stage to E4.5 during early female development.

## Embryonic *Rlim* is required for iXCI in female mice

Because RLIM protein is rapidly turned-over during early development (*Becker et al., 2003*; *Ostendorff et al., 2002*), if RLIM is required for X-silencing only after E3, this predicts that maternal stores of RLIM will not be sufficient for iXCI but that RLIM synthesized by the embryo is required for later stages of the iXCI process. To confirm that embryonic RLIM plays a crucial role for iXCI after E3 we targeted the maternally transmitted conditional *Rlim* allele via a paternally transmitted *Rosa26*-Cre (R26C) transgene (*Soriano, 1999*), from which Cre is induced after zygotic genome activation (ZGA). Indeed, an R26C-mediated deletion of the *Rlim* allele on the Xm led to embryonic lethality in a female-specific parent-of-origin effect (*Figure 4A,B*), similar to the germline *Rlim* KO (*Shin et al., 2010*). This deletion proved highly penetrant, as a maternally transmitted floxed *Rlim* allele was no longer detectable at early stages of the second iXCI phase (E3.5; *Figure 4C*). RNA FISH experiments showed that *Xist* cloud formation was strongly diminished in trophoblasts of female R26C-mediated cKO$_m$ blastocysts isolated at E4 and cultured ex vivo for 2 days (*Figure 4D*). Moreover, the detection of transcription foci of *Xist* adjacent to *Rlim* in these trophoblasts indicated defects in Xp silencing (*Figure 4E*), consistent with previously published results (*Shin et al., 2010*). These data provide genetic confirmation that embryonic RLIM expressed from the maternal allele plays crucial roles for the maintenance of *Xist* clouds during iXCI.

## *Xist* is crucial for iXCI throughout female preimplantation development

Our results show that cells of female pre/peri-implantation embryos that lack *Rlim* do not upregulate *Xist* expression and cannot maintain *Xist* clouds around stage E3.5 (*Figure 2*), and ultimately fail to silence X-linked genes at blastocyst stages (*Figure 3*). To functionally connect *Rlim* and *Xist* during iXCI we next examined female embryos that carry a germline *Xist* mutation on the paternally inherited X ($Xist^{WT/\Delta P}$). These females are devoid of functional *Xist* RNA because during iXCI, the Xp is exclusively silenced and *Xist* is not expressed from the Xm. We used a floxed *Xist* mouse line (*Csankovszki et al., 1999*) to generate males with a *Sox2*-Cre-mediated cKO of *Xist* ($Xist^{cKO/Y-SC}$). $Xist^{cKO/Y}$ males were mated with WT/WT females (*Figure 5—figure supplement 1A*) and, confirming the previously observed sex-specific parent-of-origin embryonic lethality (*Marahrens et al., 1997*), only male but no female pups were born (*Figure 5—figure supplement 1B*). Using single embryo RNA-seq, we analyzed 141 embryos generated by this cross in a similar manner as described for the RlimKO/WT dataset (*Figure 5—figure supplement 1C*; *Supplementary file 2*). Consistent with the fact that the floxed region in these mice encompasses parts of the promoter, reads in *Xist* were low in all embryos of all developmental stages (not shown). Therefore, the gender of each embryo was determined by assessing expression of Y-linked genes (*Figure 5A*). Data obtained for each embryo were processed and normalized to autosomal gene expression as described for the WT/RlimKO dataset. The global F/M expression profiles of 8127 genes in this dataset with mapped reads across all developmental stages, showed increased expression specifically of X-linked transcripts in $Xist^{WT/\Delta P}$ embryos, while general gene expression from autosomes was similar (*Figure 5B*). Confirming a central role for the *Xist* lncRNA during iXCI, the F/M expression profiles of X-linked genes were high

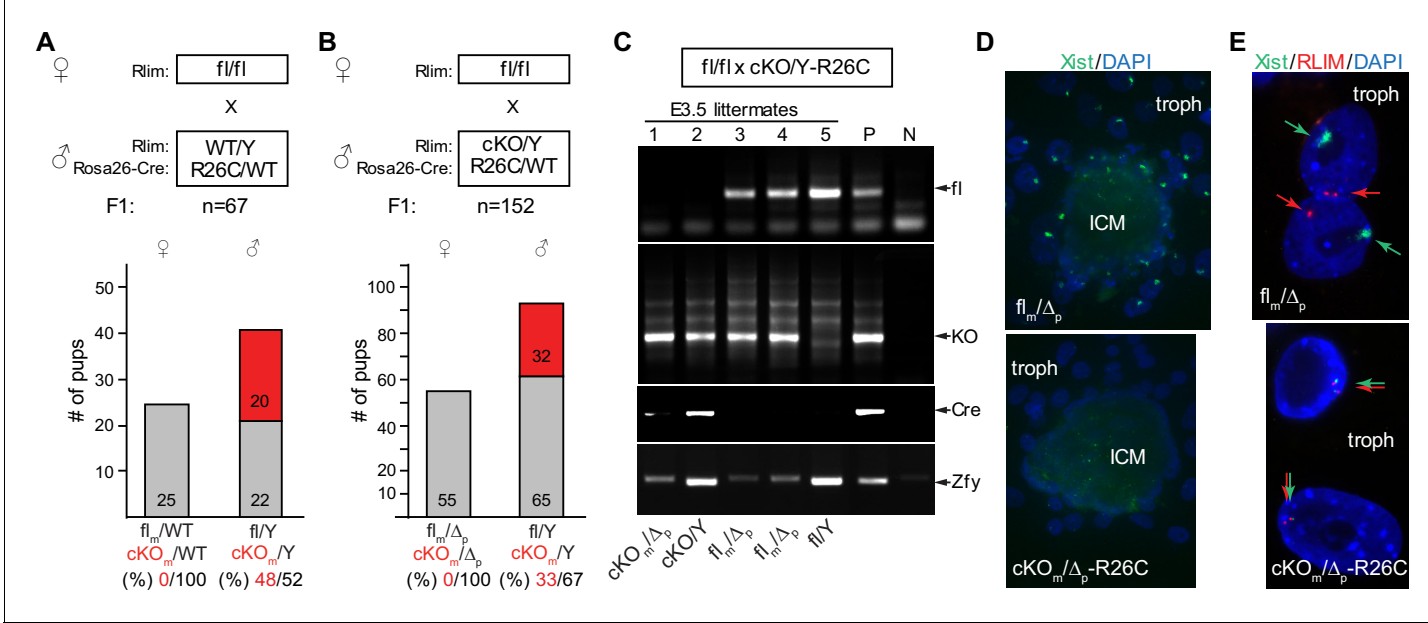

**Figure 4.** Embryonically expressed RLIM is required for iXCI in female mice. The cKO of *Rlim* in female embryos was targeted via a paternally transmitted *Rosa26-Cre* (R26C) transgene. (A, B) Schematic diagram of born pups generated via indicated mating schemes. Parental genotypes with respect to *Rlim* and *R26C* are shown. For each mating, the total number (n) of F1 offspring is indicated. Numbers of female and male pups and their genotypes with respect to *Rlim* are indicated. The *Rlim* cKOm allele in pups is indicated in red. Percentages of cKOm to fl distribution in female or male pups are shown below. Note no female offspring with an R26C-induced cKO of the maternal *Rlim* allele. (C) Robust deletion in E3.5 embryos via a paternally transmitted R26C transgene. Parental genotypes and genotypes of embryonal littermates are indicated. A slightly slower migrating band in PCRs using primers that for the Y-linked gene Zfy is unspecific. Note that the maternally transmitted floxed allele is no longer detectable in Cre-positive embryos. Positive control (P); negative control (N). The last two bands in Zfy (P; N) originate from the same gel but have been inverted to reflect the general loading pattern. (D, E) Inhibition of *Xist* clouds and X-silencing in trophoblasts of E4 female blastocyst outgrowths with an R26-Cre induced deletion of maternally transmitted *Rlim*. RNA-FISH experiments on representative *R26C-Rlim cKOm* female embryos using *Xist* (green) and *Rlim* (red) probes. Note lack of *Xist* clouds in cKO/Δ trophoblasts (D) accompanied by X-silencing defects in most trophoblast cells as indicated by side-by-side *Rlim* and *Xist* transcription foci (E). Inner cell mass, ICM; trophoblasts, troph.

in $Xist^{WT/\Delta P}$ embryos of all stages (*Figure 5C*). Comparisons with WT/WT revealed significant defects in X dosage compensation in $Xist^{WT/\Delta P}$ females at around E3.5 blastocyst stages (*Figure 5D*), consistent with published findings (*Namekawa et al., 2010*).

## Regulation of X-linked gene expression in pre/peri-implantation embryos

Studies of various adult somatic cell types have revealed general X/A expression ratios of around 1, indicating that gene expression from the active X is upregulated around two-fold. Indeed, male mouse ES cells and epiblast cells in blastocysts display X/A values of around 0.8 (*Deng et al., 2013*, *2011*, *2007*). While these results indicate incomplete X upregulation they nonetheless suggest that this upregulation might be initiated during preimplantation development. To assess alterations in the X/A ratio during preimplantation development we calculated the average total expression (FPKM) of X-linked versus total autosomal genes within each embryo. This approach removes the possibility that low expressing genes, which are more frequent on the X chromosome when compared to autosomes and vary in different cell types, influence or bias the results (*Deng et al., 2011*). Because of XCI in females, this analysis was first carried out on males. Our results revealed gradually increasing levels of transcription from the X relative to autosomes (*Figure 6A*) and the total increase between the 4-cell stage to E4.5 was 1.58 fold (P<10–11; Students t-test). Calculating X/A values using a previously published RNA-seq dataset (GSE45719 in GEO repository) on single mouse preimplantation cells (*Deng et al., 2014*), generally confirms these data (*Figure 6—figure supplement 1*). When we examined average expression from single chromosomes (normalized against 4-cell

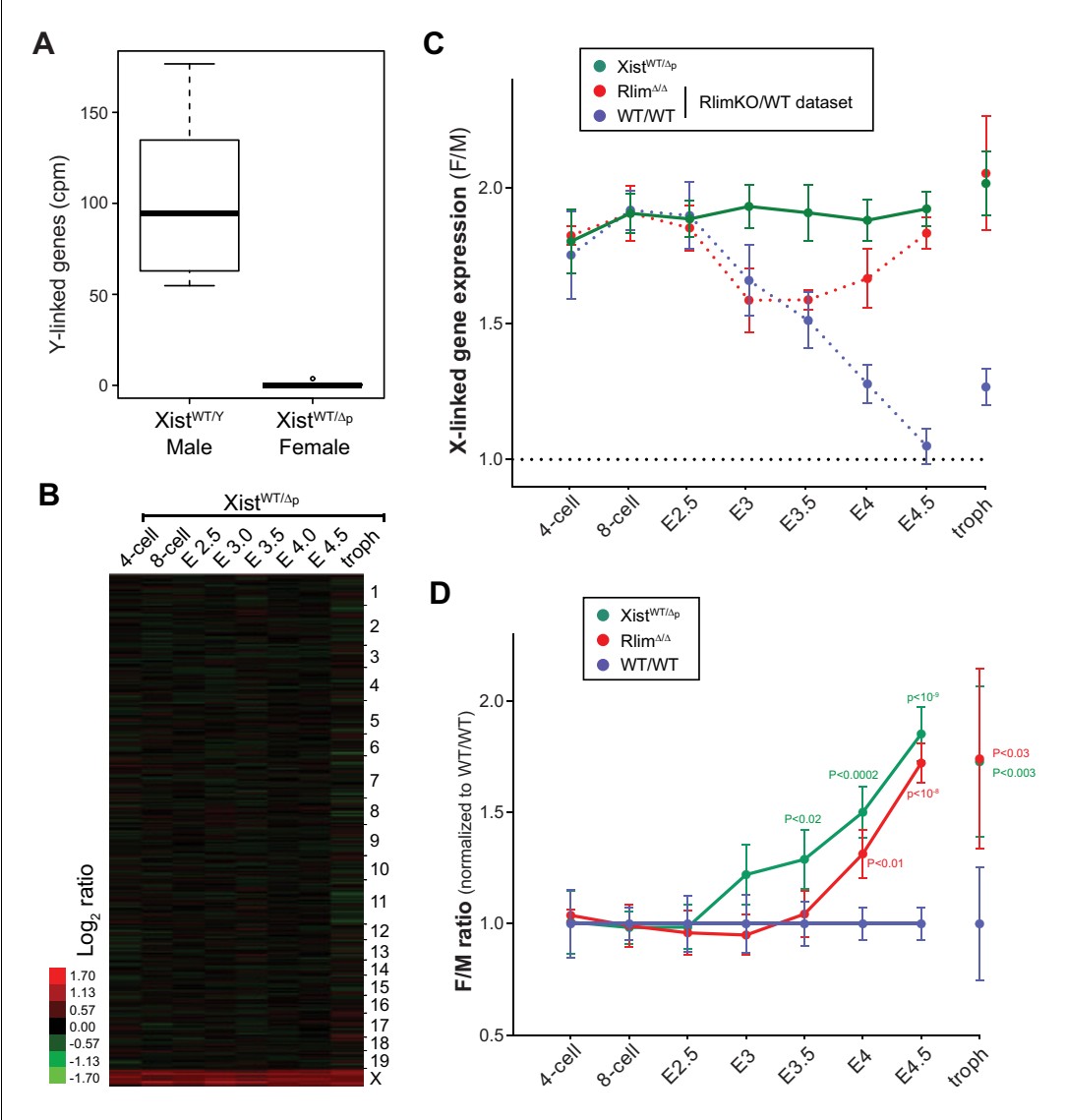

**Figure 5.** *Xist* is crucial for X dosage compensation throughout preimplantation development. All embryos were generated by crossing WT/WT females with *Xist*[Δ/Y] males. (A) Gender determination of embryos via Y-linked gene expression. As example, the distributions of reads at the 8-cell stage of Y-linked genes are shown in a box-plot. (B) Heat map representing $Log_2$ transformed data comparing the mRNA expression level ratios from chromosomes (excluding Y) between female and male embryos during pre/peri-implantation development. Chromosomes are indicated. (C) Developmental profile of X-silencing during iXCI in *Xist*[WT/Δp] females as determined by comparing F/M expression of X-linked transcripts. Data were processed as described for those obtained for the WT/RlimKO dataset which were incorporated for comparison as dotted lines (see *Figure 3A*). (D) Comparison of F/M values for *Xist*[WT/Δp] and *Rlim*[Δ/Δ] with those obtained for WT/WT (set to 1). P values of P<0.05 are indicated (paired t-test).

The following figure supplement is available for figure 5:

**Figure supplement 1.** Details of RNA-seq analyses of single embryos lacking *Xist* at pre/peri-implantation stages.

stage levels) we observed that, among all chromosomes, gene expression from the X displayed the highest increase (*Figure 6B*). Next, we compared the contribution of chromosomes towards total gene expression in males of each developmental stage by taking into account the numbers of annotated genes located on each chromosome. These analyses showed that, consistent with the presence of one X chromosome, at the earliest time point measured, the contribution of the X was much lower than that of any autosome, only around 0.5 fold of the average autosome (*Figure 6C*). This

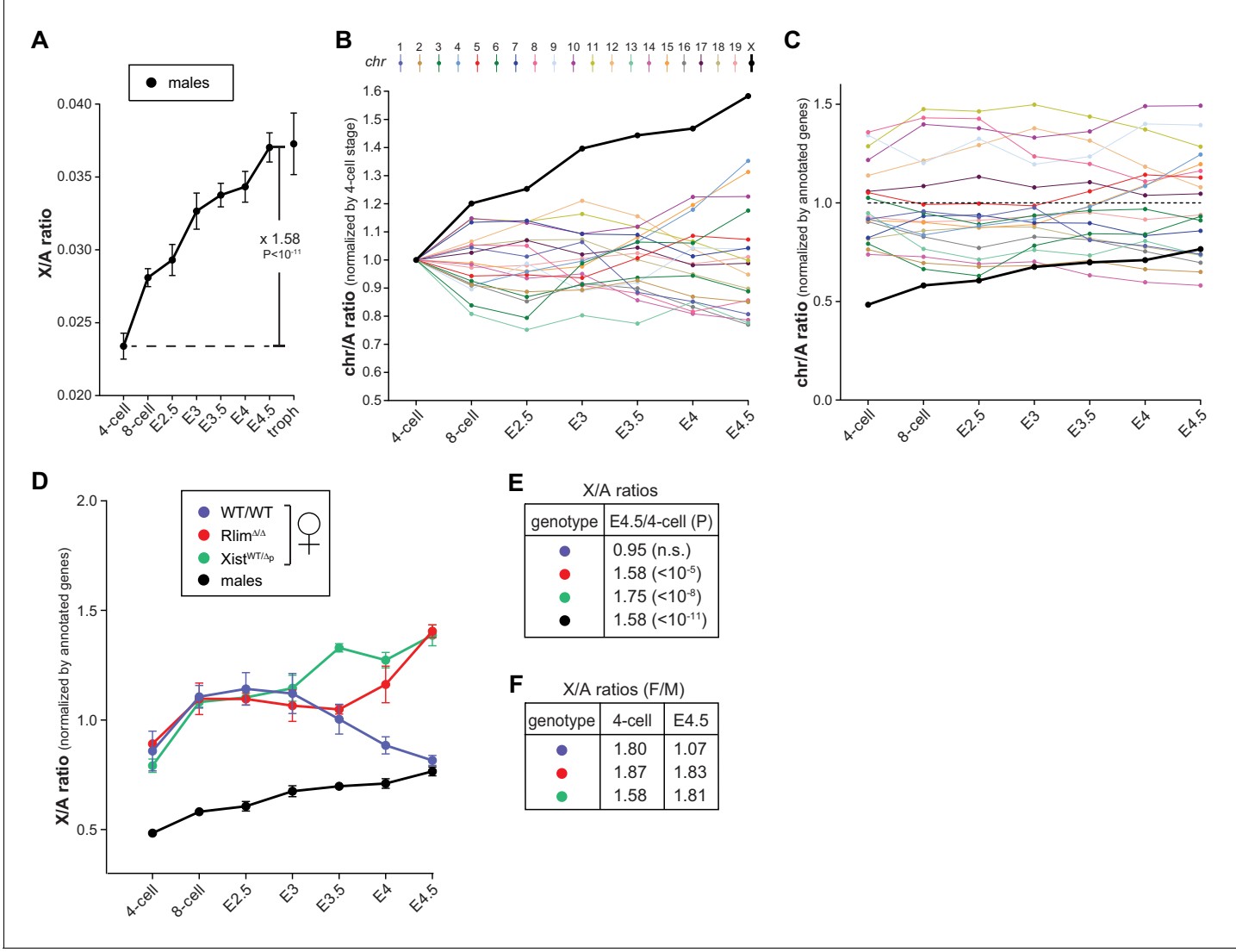

**Figure 6.** Dynamics of X-linked gene expression in preimplantation embryos. (**A**) X/A expression profile in males. Male data were collected from males of RlimKO/WT and XistKO datasets. For each embryo, the total FPKM expression of X-linked genes was divided by the total FPKM expression of autosomal genes. Shown is the average X/A values for each developmental stage. The increase of 1.58-fold in X/A values from the 4-cell stage to E4.5 is highly significant ($P<10^{-11}$; Student's t-test). (**B**) Comparison of gene expression profiles from chromosomes 1 to 19 and X during male pre/peri-implantation development. Data were collected from male embryos of both datasets. At each developmental stage the total FPKM expressed from each chromosome was divided by the total FPKM expression of all autosomal genes. Values obtained for each chromosome at the 4-cell stage are set to 1. (**C**) Dynamics of the relative gene expression expressed from single chromosomes in male embryos. The total FPKM/total number of annotated genes of each chromosome was divided by the total FPKM of all autosomal genes/total number of autosomal genes. Male data were collected from all males of both datasets. Note that expression from the X is markedly lower at early stages (around x0.5), increasing to x0.77 to that of the average autosome (1.0; dotted line) by E4.5. (**D**) Female and male X/A expression profiles normalized to annotated genes. The total FPKM/total number of annotated genes on the X was divided by the total FPKM of all autosomal genes/total number of annotated autosomal genes. Female genotypes are indicated. (**E**) The E4.5/4-cell stage ratios of X/A values are shown for WT/WT, $Xist^{WT/\Delta}$ and $Rlim^{\Delta/\Delta}$ females as well as males. P values (Student's t-test) are indicated; n.s. = not significant. (**F**) Comparison of X/A values according to gender: F/M ratios of X/A values at the 4-cell and E4.5 stages are shown for WT/WT, $Xist^{WT/\Delta}$ and $Rlim^{\Delta/\Delta}$. Error bars indicate SEM.

The following figure supplement is available for figure 6:

**Figure supplement 1.** X/A profile in male mouse embryos comparing data obtained by single embryo RNA-seq (see **Figure 6A**) with data obtained by single-cell RNA-seq (**Deng et al., 2014**).

contribution gradually increased to around 0.77 fold measured at E4.5, which was the 14th most highly expressed of the 20 chromosomes. These results are in general agreement with X/A levels previously measured in male murine ES cells as well as epiblast cells (*Lin et al., 2007*), and indicate that X upregulation is initiated during preimplantation development. Including female embryos in these analyses (*Figure 6D*), our data reveal that X/A upregulation is initiated both during male and female preimplantation development, as female $Rlim^{\Delta/\Delta}$ and $Xist^{WT/\Delta p}$ embryos, which are defective in iXCI, display high E4.5/4-cell ratios of 1.58 and 1.75, respectively, similar to those of males (*Figure 6E*). Upon examination of early developmental stages, we found that all females start out with high X/A ratios compared to those measured in males (between 1.58–1.87 at the 4-cell stage; *Figure 6F*). However, by E4.5, whereas X/A ratios of $Rlim^{\Delta/\Delta}$ and $Xist^{WT/\Delta p}$ females are still high compared to males (around 1.8 fold), those of WT/WT females have decreased to levels close to 1 (*Figure 6F*). Combined, these results indicate that X upregulation occurs during preimplantation development in males and females. In WT/WT females, X upregulation takes place concurrently with iXCI that compensates the general X gene dosage to that of males.

## Discussion

### Accuracy of single embryo RNA-seq data

Based on single cell technology, we have used single embryo RNA-seq to elucidate the mouse pre- and peri-implantation transcriptome. This approach has been recently established (*Sharma et al., 2016*) and we provide several lines of evidence for high accuracy of RNA-seq datasets at multiple levels including single genes as well as chromosome-wide: (1) Mapping reads to single genes demonstrates high fidelity of reads within the *Rlim* regions that are present on the genome (*Figure 1E*), as well as reads within the *Tsix/Xist* genomic region which allows a quantitative distinction between expression of *Xist* vs *Tsix* (*Figure 2—figure supplement 1A*). (2) The developmental expression profiles of single genes correspond to those published in the literature. This is true for cell markers (*Figure 1F*) as well as for *Xist* that is highly expressed in females but not males (*Figure 2A*). (3) Our results obtained by RNA-seq on *Xist* expression (*Figure 2A*) have been confirmed using alternative methods such as strand-specific RT-qPCR (*Figure 2—figure supplement 1C*) and RNA-FISH (*Figure 2B,C*). Analyzing chromosome-wide gene expression of WT mouse embryos reveals that the X dosage in females is around two-fold that of males at early embryonic stages and is subsequently adjusted to those in males at late stages (*Figure 3A*), in agreement with recent findings in mouse and human embryos (*Petropoulos et al., 2016*; *Chen et al., 2016*). Analysis of XistKO females that cannot undergo iXCI shows, as expected, around two-fold higher X-linked gene dosages when compared to males throughout preimplantation development. These data combined with results obtained by subsampling of library sizes (*Figure 1—figure supplement 1E*; *Figure 2—figure supplement 1B*; *Figure 3—figure supplement 1A,B*) and direct comparisons of various genotypes (WT vs RlimKO; female and male; e.g. *Figure 3—figure supplements 1C, D; 2*; not shown) confirm the high overall robustness and accuracy of our RNA-seq data. However, likely reflecting technical variability of single-embryo RNA-seq, variations of single gene expression levels among replicates were generally higher when compared to those of chromosome/genome-wide data, where hundreds/thousands of genes are averaged.

### Chromosome-wide X-linked gene expression

While crucial functions of *Rlim* and *Xist* for iXCI in mice are known (*Marahrens et al., 1997*; *Shin et al., 2010*), the influence of both genes on the general kinetics of X-linked gene expression are not. To study iXCI, X upregulation and the effects of *Xist* and *Rlim* on X-linked gene expression we have used embryos with congenic genetic background because this approach allows for simpler mouse genetics and excludes influences of the genetic background on X-linked gene expression. The facts that (1) females have two and males have one X chromosome and (2) *Xist* is crucial for iXCI, comparing females to males combined with the inclusion of the XistKO mouse model allows for an overall chromosome-wide assessment of the dynamics of X-linked gene expression via comparisons of gene expression profiles between the genders and/or between females with different genotypes. This is because XistKO female embryos display around two-fold higher F/M levels throughout early development (*Figure 5A*), thereby providing genetic evidence that the global X profile

differences between XistKO and WT/WT females is the direct consequence of iXCI and that there is no other, major female-specific mechanism in preimplantation embryos that significantly influences the global X-linked gene dosage. Concerning allele-specific X-chromosome-wide gene expression, as males have a single maternally inherited X combined with the fact that iXCI in females silences exclusively the Xp (*Deng et al., 2014*), indicates that the steady decline in F/M values seen in WT/WT females is mostly due to decreased Xp expression levels, and using $B6_m/CAST_p$ hybrid embryos we have confirmed this for stage E3.5 (*Figure 3—figure supplement 2*). However, in contrast to chromosome-wide X-linked gene expression, at the gene level we cannot resolve the parental origin of single RNAs, as our RNA-seq data do not reveal an allele-specific resolution.

In agreement with published results (*Deng et al., 2014*; *Marks et al., 2015*), we show that silencing of X-linked genes in females occurs across the entire X chromosome (*Figure 3C*). Such spatial concordance of silencing is consistent with studies in ESCs which show that *Xist* does not spread linearly along the X chromosome, but rather spreads to multiple loci on the X simultaneously, due to the three dimensional folding of the chromosome (*Engreitz et al., 2013*). The finding that the average F/M ratios of females of all genotypes (WT/WT, $Rlim^{\Delta/\Delta}$ and $Xist^{WT/\Delta p}$) remained below 2 even at early developmental stages (*Figures 2A; 5C*) is likely explained by a previously reported *Xist*-independent partial silencing of some X-linked transcripts (*Kalantry et al., 2009*). Thus, by integrating X upregulation and iXCI, our results represent a comprehensive view on X-linked gene expression in early mouse embryos, and its regulation by *Rlim* and *Xist*.

Our results reveal that the general pre/peri-implantation profile of X dosage compensation between genders as observed in WT female mice (*Figure 3A*) is remarkably similar to that measured in early female human embryos (*Petropoulos et al., 2016*), even though mice but not humans undergo iXCI (*Okamoto et al., 2011*). This suggests strong evolutionary pressure on X dosage compensation before implantation. It is thus surprising that unlike in mice (*Figure 6*) there is no sign of X upregulation in human preimplantation embryos (*Petropoulos et al., 2016*) (not shown). Because X/A values in adult human somatic tissues are close to 1 (*Deng et al., 2011*), this suggests that X-linked gene expression is upregulated at post-implantation stages.

## Regulation of iXCI by *Rlim*

Our analyses of global gene expression profiles in females shows that the KO of *Rlim* or *Xist* affects global expression levels of X-linked transcripts but not those expressed from autosomes (*Figures 3B; 5B*). This combined with the fact that males lacking either *Xist* or *Rlim* appear healthy and are fertile indicates that crucial roles for both genes are restricted to X dosage compensation in females. Indeed, our results reveal that high levels of *Xist*, the maintenance of *Xist* clouds and X-dosage compensation in female blastocysts depend on *Rlim*. Combined, these results imply that RLIM's function on iXCI is exerted through regulation of *Xist*. Moreover, because only iXCI globally and significantly influences X-linked gene expression specifically in females, the development of temporary *Xist* clouds in RlimKO females (*Figure 2*) and a similar silencing pattern at E3 between WT/WT and RlimKO females (*Figure 3*; *Figure 3—figure supplement 1C*) suggests that initiation of the iXCI process might occur normally at early stages in RlimKO females. However, at blastocyst stages X-silencing and iXCI cannot be maintained leading to a failure in X dosage compensation (*Figure 3*; *Figure 3—figure supplement 1D*). This is further corroborated by our results targeting the *Rlim* cKO via Rosa26-Cre early during female embryogenesis (*Figure 4*), confirming crucial roles for the maintenance of iXCI in blastocyst-staged female embryos. In this context, it is important to note that RLIM protein levels are down-regulated specifically in epiblast cells of E4.5 blastocyst embryos (*Shin et al., 2014*), at a time point when these cells start to reactivate the Xp (*Mak et al., 2004*; *Okamoto et al., 2004*) before undergoing rXCI. Thus, because RLIM is crucial for the maintenance of iXCI at peri-implantation stages (*Figures 2–4*), this down-regulation provides an attractive molecular mechanism for triggering the Xp reactivation process.

In summary, by elucidating the mouse pre- and peri-implantation transcriptome, this study provides a comprehensive view on X linked gene expression during early mouse development. Our analyses uncover that upregulation of X-linked transcripts is initiated in early male and female mouse embryos. In females X upregulation occurs concomitant with iXCI, which progressively leads to X gene dosage compensation between genders in an *Rlim* and *Xist*-dependent manner.

## Materials and methods

### Mice

Mice used in this study and genotyping have been described; *Rlim fl/fl* (*Shin et al., 2010*), *Xist fl/fl* (*Csankovszki et al., 1999*), *Sox2*-Cre (*Hayashi et al., 2002*; *Shin et al., 2014*) and *Rosa26-Cre* (*Soriano, 1999*). The *Xist^{fl/fl}* mouse strain 129-Xisttm2Jae/Mmnc, identification number 29172-UNC, was obtained from the Mutant Mouse Regional Resource Center, a NIH funded strain repository, and was donated to the MMRRC by Rudolf Jaenisch, Ph.D., Whitehead Institute. CAST/Eij mice were purchased from The Jackson Laboratories. Rlim^{fl/fl} mice were generally bred and maintained on a C57BL/6 background. All mice were housed in the animal facility of UMMS, and utilized according to NIH guidelines and those established by the UMMS Institute of Animal Care and Usage Committee.

### Single-embryo RNA-seq

All embryos were generated by natural mating. Whole embryos were dissected at the indicated time points and the correct stage was verified under the binocular. Single-Embryo RNA-seq was essentially performed as described (*Sharma et al., 2016*). Briefly, single embryos/trophoblast cells were placed in 10ul TCL Buffer (Qiagen) supplemented with 1% BME, and snap frozen. A total of 187 samples representing WT and *Rlim*KO embryos were distributed on two 96-well plates (plus 5 mock wells) and thawed at RT for 10 min prior to RNA purification using Ampure RNA beads (Beckman-Coulter, Brea, CA). RNA samples were resuspended in solution containing 3' RT primer (5'-AAGCAGTGGTATCAACGCAGAGTACT(30)VN-3') and dNTPs. Reverse transcription was performed with SSII (Life Technologies), whose terminal transferase activity allows incorporation of a PCR binding site at the 3' end of the cDNA using a template-switching oligonucleotide (5'-AAGCAGTGGTATCAACGCAGAGTACATrGrG G-3'; custom synthesized from Exiqon) as a template. Subsequently, cDNA was amplified using 12 cycles of PCR, followed by tagmentation with Nextera kit (Illumina). Final libraries were amplified by 12 cycles of PCR (5'-AAGCAGTGGTATCAACGCAGAGT-3') and sequenced on a NextSeq 500. Single-embryo RNA-seq of *Xist*KO or B6/CAST hybrid embryos was performed as described above.

### RNA-seq data analyses

Reads (paired end 35 bp) were aligned to the mouse genome (mm10) using TopHat (version 2.0.12) (*Trapnell et al., 2009*), with default setting except set parameter read-mismatches to 2, followed by running HTSeq (version 0.6.1p1) (*Anders et al., 2015*), Bioconductor packages edgeR (version 3.10.0 ) (*Robinson et al., 2010*; *Robinson and Smyth, 2007*) and ChIPpeakAnno (version 3.2.0) (*Zhu, 2013*, *2010*) for transcriptome quantification, differential gene expression analysis, and annotation. For edgeR, we followed the workflow as described in (*Anders et al., 2013*), except that the library size of each embryo was set as the total number of the effective counts of the autosomal genes. Specifically, edgeR was used for the removal of the unmapped, ambiguous, and not annotated reads as well as reads in rRNA and the filtering out of low expression genes after regrouping samples according to developmental stage, as the X chromosome contains many reproduction-related genes that are not expressed in somatic tissues (*Khil et al., 2004*; *Mueller et al., 2008*) and genes with no or low expression may influence the X/A expression ratios (*Deng et al., 2011*). Therefore, genes that were not expressed or lowly expressed genes as determined for embryos at each stage were filtered out before normalization, via removal of genes without at least 1 read per million in n of the samples, where n is the size of the smallest group of biological replicates within each developmental stage (*Anders et al., 2013*). The library size was then reset as the total number of the effective counts of autosomal genes. The TMM method (Trimmed Mean of M-value) was used to calculate normalization factors between samples of the same stage (*Robinson and Oshlack, 2010*). Fragments per kilobase of exon per million reads mapped (FPKM) (*Supplementary files 1*, *2*) and LogFC were calculated using edgeR. For gender assessment counts per million (cpm) of mapped reads in *Xist* and the seven Y-linked genes *Ddx3y*, *Eif2s3y*, *Kdm5d*, *Usp9y*, *Uty*, *Zfy1* and *Zfy2* were evaluated. 12 samples out of 187 sequenced in the combined *Rlim*KO/WT RNA-seq experiment, and 33 samples out of 174 sequenced in the *Xist*KO RNA-seq experiment were disregarded due to low reads or because gender could not be clearly determined. Random subsampling of library sizes to 200.000 reads per embryo was performed as described (*Robinson and Storey, 2014*). Analyses of

the subsampled datasets were carried out as described above. F/M analyses were carried out by averaging ratios per gene within each developmental stage using females of defined genotype and pooled males (WT and KO). For calculations in *Figure 3—figure supplement 1C,D*, 755 X-linked genes with cpms>1 (before normalization) in at least three embryos at each stage were included. The dataset was then normalized against autosomal gene expression and Log2 fold change was calculated using edgeR. For the allele-specific expression analysis of X-linked transcripts in B6/CAST heterozygous females, the SNPs were called with mpileup and bcftools in the SAMtools package (*Li, 2011*) using the aligned BAM files. A python program allelecounter was used to obtain the allele counts (https://github.com/secastel/allelecounter). SNPs called in females were verified by comparisons to sequences in males (C57BL/6). Data analysis was carried out by comparing SNPs with reference genomes C57BL/6 (mm10; UCSC) and CAST/Eij (http://csbio.unc.edu/CCstatus/index.py?run= Pseudo).

## Blastocyst outgrowths, RNA fluorescence in situ hybridization (RNA FISH), Immunohistochemistry and RT-qPCR

All embryos were generated by natural mating and harvested at the indicated embryonic stages. For blastocyst outgrowths, embryos were harvested at E4, cultured for 48 hr and genotyped after image recording. RNA FISH was performed essentially as previously reported (*Shin et al., 2010*; *Byron et al., 2013*). For the synthesis of specific *Xist* probes, we used plasmids containing mouse *Xist* exon 1 and 6 that recognize *Xist* and *Tsix* (*Panning, 2004*). For the *Rlim* probe, we used a plasmid containing genomic *Rlim* sequences upstream of the KO site that detects specific *Rlim* mRNAs transcribed from both wild type and KO alleles (*Shin et al., 2010*). Ovaries of three 8-weeks old WT/WT and *SC-Rlim*$^{cKO/\Delta}$ females each were dissected, fixed and stained with an RLIM antibody as described previously (*Shin et al., 2010*). RT-qPCR on whole embryos using primers that detect RNA transcribed from *Xist* and actin as control, were performed as previously reported (*Shin et al., 2010*).

## Acknowledgements

We thank T Fazzio, L Hall and S Jones for reading of the manuscript and/or advice and discussion and S Kalantry for the *Rosa26-Cre* mice. RNA-seq data have been deposited with the GEO repository (GSE71442). IB is a member of the University of Massachusetts DERC (DK32520). This work was supported from NIH grants R01CA131158 to IB, R01HD080224 and DP1ES025458 to OJR, and R01GM053234 to JBL.

## Additional information

### Competing interests

AR: Senior editor, *eLife*. The other authors declare that no competing interests exist.

### Funding

| Funder | Grant reference number | Author |
| --- | --- | --- |
| National Institutes of Health | R01CA131158 | Ingolf Bach |
| National Institutes of Health | R01HD080224 | Oliver J Rando |
| National Institutes of Health | DP1ES025458 | Oliver J Rando |
| National Institutes of Health | R01GM053234 | Jeanne B Lawrence |

The funders had no role in study design, data collection and interpretation, or the decision to submit the work for publication.

### Author contributions

FW, JDS, Conception and design, Acquisition of data, Analysis and interpretation of data; JMS, Designed experiments, Acquisition of data; JY, EMT, LJZ, Analyzed RNA-seq datasets, Analysis and interpretation of data; AB, AKS, OJR, Carried out the RNA seq on single embryos, Acquisition of data; MB, XZ, JBL, Carried out the RNA FISH experiments, Acquisition of data; AR, Acquisition of

data; IB, Conception and design, Acquisition of data, Analysis and interpretation of data, Drafting or revising the article

## Author ORCIDs

Ingolf Bach, http://orcid.org/0000-0003-4505-8946

## Ethics

Animal experimentation: All mice were housed in the animal facility of UMMS, and utilized according to NIH guidelines and those established by the UMMS Institute of Animal Care and Usage Committee (IACUC protocol #: A-1940-14).

# Additional files

## Supplementary files

• Supplementary file 1. RlimKO/WT dataset: List of FPKM for all embryos at all stages (4-cell, 8-cell, E2.5, E3, E3.5, E4, E4.5 and troph).

• Supplementary file 2. XistKO dataset: List of FPKM for all embryos at all stages (4-cell, 8-cell, E2.5, E3, E3.5, E4, E4.5 and troph).

## Major datasets

The following dataset was generated:

| Author(s) | Year | Dataset title | Dataset URL | Database, license, and accessibility information |
|---|---|---|---|---|
| Bach I | 2015 | Transcriptome of mouse preimplantation development | http://www.ncbi.nlm.nih.gov/geo/query/acc.cgi?acc=GSE71442 | Publicly available at the NCBI Gene Expression Omnibus (accession no: GSE71442) |

The following previously published dataset was used:

| Author(s) | Year | Dataset title | Dataset URL | Database, license, and accessibility information |
|---|---|---|---|---|
| Deng Q, Ramsköld D, Reinius B, Sandberg R | 2014 | Single-cell RNA-Seq reveals dynamic, random monoallelic gene expression in mammalian cells | http://www.ncbi.nlm.nih.gov/geo/query/acc.cgi?acc=GSE45719 | Publicly available at the NCBI Gene Expression Omnibus (accession no: GSE45719) |

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
