## [Decision Letter]

Thank you for submitting your article "Dynamics of X-linked gene expression during early mouse development" for consideration by *eLife*. Your article has been reviewed by two peer reviewers, and the evaluation has been overseen by Kevin Struhl as the Senior Editor and the Reviewing Editor. The reviewers have opted to remain anonymous.

The reviewers have discussed the reviews with one another and the Reviewing Editor has drafted this decision to help you prepare a revised submission.

This paper uses RNA-seq on single mouse embryos to elucidate thepre/peri-implantation transcriptome. It shows that iXCI in females globally compensates the X gene dosage with crucial roles of Rlim for increased Xist levels and maintenance of iXCI at blastocyst stages and that X/A upregulation is initiated in early male and female embryos.

The reviewers were very favorable about the RLIM part of the work and believe that this should be the focus of the paper. They appreciated the RNA-seq experiments on embryos as opposed to what has been done previously in ES cells. However, they felt that single-cell RNA-seq analysis of embryos was critical for making conclusions about the dynamics of X-linked expression. Without this, the results on embryos were viewed as only an incremental advance over published work. In particular Chen et al. (Genome Research online) "Single-cell analyses of X Chromosome inactivation dynamics and pluripotency during differentiation" follow up on the Deng et al. (Science, 2014 "Single-cell RNA-seq reveals dynamic, random monoallelic gene expression in mammalian cells) study, so single cell early development RNA-seq exists for mouse (as well as human – Petropoulos).

Thus, a revised manuscript would be accepted if it were significantly rewritten to focus on RLIM, including changes in the title, Abstract, and throughout the text. Single-cell RNA-seq experiments on embryos are not required. However, if they are not done (and presumably this would take considerable time), the "dynamics" aspect of the work must be considerably downplayed including citing all the relevant work on the subject and placing the current results into that context. To put it simply, the RLIM aspect of the work is of sufficient interest for *eLife*, whereas the dynamics aspect is not. The original reviews are included below, but the summary above comes from considerable discussion among the reviewers and BRE.

*Reviewer #1:*

Wang et al. report a survey of the pre- and peri-implantation transcriptome using RNA-seq for single mouse embryos to compare wild-type and knock-outs for the Rlim and Xist genes. These genes have been previously implicated in the process of X inactivation. The mammalian X chromosome is regulated by two major mechanisms that evolved due to the fundamental difference between males (XY) and females (XX). These mechanisms restore a balanced expression between X and autosomes (X upregulation) and between the sexes (X inactivation).

The data obtained here, combined with RNA-FISH, provide a comprehensive view of the dynamics of X-linked gene expression during early embryo development and elucidates a crucial role for the maternal and embryonic RLIM protein in upregulation/maintenance of Xist expression, which is essential for the completion of imprinted X inactivation. This study also clearly shows that X upregulation initiates in both male and female mouse preimplantation embryos, supporting the idea of X upregulation occurs in both sexes to balance expression between the single active X and the autosomes present in two copies. Overall, the conclusions are strongly supported by the data and the manuscript is well written. While the changes in X-linked gene expression during the onset of X inactivation in ES cell differentiation were already known this study extends the findings to mouse embryos.

A few points need to be clarified:

1) The authors should explain in the Methods how the female/male expression ratios of X-linked genes were calculated. Why do they use FPKM in this analysis instead of cpm that is used for expression of Y-linked genes, Xist, Oct4, Nanog, Krt8 and Rlim (Figure 1)? In addition, FPKM and cpm should be annotated.

2) In Figure 3, F/M expression profiles of 351 instead of 755 X-linked genes were used. The authors need to explain how these 351 genes were chosen.

3) In the Introduction, the role of RLIM in random XCI should also be mentioned.

4) The citations at the beginning of the first paragraph of the Results are not all appropriate since the work of Yang et al. 2010 is about mouse fibroblasts and the paper by Berletch et al. 2010 is an early review that does not mention mouse ES cell work. In addition to the Marks paper the authors should also mention the paper by Lin et al., 2007 who showed very similar data in ES cells as those shown here in an in vivo system, and the paper by Calabrese et al., 2012, who used RNA-seq to follow imprinted X inactivation in trophoblastic cells.

5) Figure 2—figure supplement 1 would be better if fully included in Figure 2 including RNA-FISH pictures. The "boxed area" mentioned in Figure 2 was not visible.

*Reviewer #2:*

This manuscript on the 'Dynamics of X-linked gene expression during early mouse development' focusses on the role of RNF12 in the initial regulation of global X-linked gene expression in females by using single mouse embryo RNA-seq.

The study of the initial X inactivation process is challenging as there is cell differentiation and global X up-regulation occurring at the same, or close to the same, time. As well the initial inactivation in mice is imprinted, and maintained in the trophoblast while being reset in epiblast cells.

The authors approach the question using Rlim-KO and Xist-KO embryos and utilize whole embryo sequencing. While they briefly discuss the advantage of a pure strain over a cross that would allow allele-specific assessment and the validity of whole embryo over single-cell sequencing, I am not convinced that this was the most informative approach. They base their 'dynamics' on the assumption that there is only iXCI occurring in their cells, that few genes escape XCI and that a mix of three cell types is not problematic. In addition, they assume that Rlim is only important in females. These assumptions limit the assessment of 'dynamics' as indicated in the title, and to me mean that the study is predominantly of the impact of RNF12, which while still interesting, may not be of broad enough interest for *eLife*. I would also like to see them distinguish which previous studies used which approaches. I think there is considerable merit in examining early embryos rather than differentiating ES cells, which is what is done in some of the previous studies.

Overall the results show that Rlim is dispensible for initial (4cell) activation of Xist, but essential to maintain the Xist cloud. However, the role for Rlim in Xist maintenance is not revealed. Similarities are cited to human X inactivation; however substantial changes in timing are not noted.

Data:

I found the presentation somewhat challenging to follow, with Figures cited out of order, and supplemental data.

Figure 1: Validation of Rlim deletion and embryo sexing

In Figure 1 why does the downstream region of Rlim vary so much – between WT and deletion, and also between WT floxed region and between developmental timepoints? I found it odd that Rlim was only assessed in females (Figure 1) – does it differ in male? GTEX suggests male and female human levels are similar, however it has been reported to be mutated in human intellectual disability. (Figure 1—figure supplements 2B and C are before Figure 2 and Figure 3 show the data).

Figure 2: Impact of Rlim KO on Xist expression

Rlim deletion shows clear downregulation of Xist after E3.5 (or even 8-cell). I think it would be useful to have the counts (i.e. Figure 2—figure supplement 1) shown on Figure 2.

Figure 3: Impact on X-linked gene expression.

It would be helpful to indicate that the y axes in C are log2 F/M ratios.

When examining the X-linked gene expression (F/M; e.g. Figure 1—figure supplement 2C; or Figure 3) is the WT and KO F compared to WT and KO M? Also, cpms is not defined – how does reads per million differ from (cpm) (in legends) and also from FPKM. I presume the ratios are done per gene and then averaged?

Figure 4: Requirement for embryonic Rlim

Figure 5: Differences between Xist ko and Rlim ko

At this point I was comparing Figure 3 (and 2) to Figure 5, and would have liked to see the direct comparison that is presented in Figure 5 also include the expression levels of Xist. To me, panel D was not helpful, although it did give p values.

Figure 6: X/A ratios in the Xist and Rlim mutants become similar.

It was nice to see all the autosomes plotted, although I question whether panels B and C are both necessary.

*Reviewer #2 (Additional data files and statistical comments):*

The.xls of the processed data are a useful addition to access through GEO.

---

## [Author Response]

*Reviewer #1:*

Wang et al. report a survey of the pre- and peri-implantation transcriptome using RNA-seq for single mouse embryos to compare wild-type and knock-outs for the Rlim and Xist genes. These genes have been previously implicated in the process of X inactivation. The mammalian X chromosome is regulated by two major mechanisms that evolved due to the fundamental difference between males (XY) and females (XX). These mechanisms restore a balanced expression between X and autosomes (X upregulation) and between the sexes (X inactivation).

*The data obtained here, combined with RNA-FISH, provide a comprehensive view of the dynamics of X-linked gene expression during early embryo development and elucidates a crucial role for the maternal and embryonic RLIM protein in upregulation/maintenance of Xist expression, which is essential for the completion of imprinted X inactivation. This study also clearly shows that X upregulation initiates in both male and female mouse preimplantation embryos, supporting the idea of X upregulation occurs in both sexes to balance expression between the single active X and the autosomes present in two copies. Overall, the conclusions are strongly supported by the data and the manuscript is well written. While the changes in X-linked gene expression during the onset of X inactivation in ES cell differentiation were already known this study extends the findings to mouse embryos.*

*A few points need to be clarified:*

*1) The authors should explain in the Methods how the female/male expression ratios of X-linked genes were calculated.*

F/M analyses were carried out by averaging ratios per gene within each developmental stage. We have added a brief quick description in the Materials and methods section as requested.

*Why do they use FPKM in this analysis instead of cpm that is used for expression of Y-linked genes, Xist, Oct4, Nanog, Krt8 and Rlim (Figure 1)?*

We used cpm for the analyses of expression of single genes as well as analyses of few genes including sex determination purposes where expression of the seven Y-linked genes *Ddx3y, Eif2s3y, Kdm5d, Usp9y, Uty, Zfy1* and *Zfy2* were evaluated (see Materials and methods section). For analyzing/comparing expression of many genes (e.g. F/M) we included normalization for transcript lengths in our expression analyses by calculating and comparing FPKM values.

*In addition, FPKM and cpm should be annotated.*

We have annotated FPKM and cpm in the Materials and methods section and in Figure legends to Figure 1 and Figure 3.

*2) In Figure 3, F/M expression profiles of 351 instead of 755 X-linked genes were used. The authors need to explain how these 351 genes were chosen.*

This number reflects the number of X-linked genes left after filtering out low expression genes (threshold: genes without at least 1 read per million in n of the samples, where n is the size of the smallest group of replicates within each stage). See data processing as described in the Materials and methods section.

*3) In the Introduction, the role of RLIM in random XCI should also be mentioned.*

We have briefly mentioned the role of RLIM in random XCI in the Introduction. Only briefly, because we feel that it is important to keep this paper focused on iXCI.

*4) The citations at the beginning of the first paragraph of the Results are not all appropriate since the work of Yang et al. 2010 is about mouse fibroblasts and the paper by Berletch et al. 2010 is an early review that does not mention mouse ES cell work. In addition to the Marks paper the authors should also mention the paper by Lin et al., 2007 who showed very similar data in ES cells as those shown here in an* in vivo *system, and the paper by Calabrese et al., 2012, who used RNA-seq to follow imprinted X inactivation in trophoblastic cells.*

We have corrected the references as requested (Results, first paragraph).

*5) Figure 2—figure supplement 1 would be better if fully included in Figure 2 including RNA-FISH pictures. The "boxed area" mentioned in Figure 2 was not visible.*

We have incorporated this figure as Figure 2. The boxed area is visible in our pdf file version of Figure 2. If this continues to be a problem please let us know and we will supply a Figure with increased box stroke.

*Reviewer #2:*

*This manuscript on the 'Dynamics of X-linked gene expression during early mouse development' focusses on the role of RNF12 in the initial regulation of global X-linked gene expression in females by using single mouse embryo RNA-seq.*

*The study of the initial X inactivation process is challenging as there is cell differentiation and global X up-regulation occurring at the same, or close to the same, time. As well the initial inactivation in mice is imprinted, and maintained in the trophoblast while being reset in epiblast cells.*

*The authors approach the question using Rlim-KO and Xist-KO embryos and utilize whole embryo sequencing. While they briefly discuss the advantage of a pure strain over a cross that would allow allele-specific assessment and the validity of whole embryo over single-cell sequencing, I am not convinced that this was the most informative approach. They base their 'dynamics' on the assumption that there is only iXCI occurring in their cells, that few genes escape XCI and that a mix of three cell types is not problematic. In addition, they assume that Rlim is only important in females. These assumptions limit the assessment of 'dynamics' as indicated in the title, and to me mean that the study is predominantly of the impact of RNF12, which while still interesting, may not be of broad enough interest for eLife. I would also like to see them distinguish which previous studies used which approaches. I think there is considerable merit in examining early embryos rather than differentiating ES cells, which is what is done in some of the previous studies.*

We have added references describing investigations on X-silencing (Results, first paragraph) as well as studies using single cell RNA-seq investigating human and mouse preimplantation embryos (subsection “Rlim regulates X-silencing during blastocyst stages”, first paragraph).

*Overall the results show that Rlim is dispensible for initial (4cell) activation of Xist, but essential to maintain the Xist cloud. However, the role for Rlim in Xist maintenance is not revealed. Similarities are cited to human X inactivation; however substantial changes in timing are not noted.*

In response to this comment we have changed the wording and replaced “general kinetics of X dosage compensation […] is similar (between mouse and human)” with “general pre/peri- implantation profile of X dosage compensation […] is similar” (subsection “Chromosome-wide X-linked gene expression”, last paragraph).

*Data:*

*I found the presentation somewhat challenging to follow, with Figures cited out of order, and supplemental data.*

In response to this comment, we have adjusted Figures to the order of data appearance in the text and distributed the results on subsampling of RNA seq data (previously summarized in a single Figure 1—figure supplement 2A-D) in figure supplements to Figure 1, Figure 2 and Figure 3.

*Figure 1: Validation of Rlim deletion and embryo sexing*

*In Figure 1 why does the downstream region of Rlim vary so much – between WT and deletion, and also between WT floxed region and between developmental timepoints?*

In Figure 1 (and Figure 2—figure supplement 1) what is shown are raw reads *before* normalization. Thus, the scale reads are represented is variable between WT and KO and between stages. In response to this comment we have indicated in Figure legends that reads indicate raw reads.

*I found it odd that Rlim was only assessed in females (Figure 1) – does it differ in male? GTEX suggests male and female human levels are similar, however it has been reported to be mutated in human intellectual disability. (Figure 1—figure supplements 2B and C are before Figure 2 and Figure 3 show the data).*

Rlim is X-linked and therefore is expressed from one and two alleles in males and females, respectively. Because Rlim regulates iXCI in females we have focused on Rlim expression in females only.

*Figure 2: Impact of Rlim KO on Xist expression*

*Rlim deletion shows clear downregulation of Xist after E3.5 (or even 8-cell). I think it would be useful to have the counts (i.e. Figure 2—figure supplement 1) shown on Figure 2.*

See comment 5 of reviewer 1):We have incorporated Figure 2—figure supplement 1 as Figure 2.

Figure 3: Impact on X-linked gene expression.

*It would be helpful to indicate that the y axes in C are log2 F/M ratios.*

We have changed the Figure legends to clearly indicate this.

*When examining the X-linked gene expression (F/M; e.g. Figure 1—figure supplement 2C; or Figure 3) is the WT and KO F compared to WT and KO M? Also, cpms is not defined – how does reads per million differ from (cpm) (in legends) and also from FPKM. I presume the ratios are done per gene and then averaged?*

F/M analyses were carried out by averaging ratios per gene within each developmental stage using females of defined genotype and pooled males (WT and KO). This is indicated in the Materials and methods. We have defined cpms and FPKM (see our response to Comment 1 of reviewer 1).

*Figure 5: Differences between Xist ko and Rlim ko*

*At this point I was comparing Figure 3) to Figure 5, and would have liked to see the direct comparison that is presented in Figure 5 also include the expression levels of Xist.*

We feel that adding Xist levels of female WT/WT and RlimKO embryos (data from Figure 3) to the F/M profiles of WT, RlimKO and XistKO animals would make this proposed figure highly confusing. This is because in the proposed figure there would be five different curves, representing different measures (two Y axes) and there are relatively large variations in Xist levels within embryos of each stage (large error bars), adding to the confusion. Because the developmental time points in Figure 2 and Figure 5 are the same, profile comparisons can be made relatively easily even if data originate from separate Figures. Thus, we feel that this proposed merged Figure would not be productive.

*To me, panel D was not helpful, although it did give p values.*

We believe that P-values are important to show also for the XistKO F/M data (compared to WT/WT F/M data). But as panel 5C represents an overlay of data from two independent datasets (RlimKO/WT and XistKO) and there are no WT/WT females within the XistKO dataset, P values cannot be calculated.

Figure 6: X/A ratios in the Xist and Rlim mutants become similar.

*It was nice to see all the autosomes plotted, although I question whether panels B and C are both necessary.*

Panel B shows the developmental profile of transcription from each chromosome, indicating that upregulation occurs specifically of X-linked transcripts during the recorded developmental stages. Panel C shows the contribution of X-linked gene expression towards total gene expression compared to each autosome. The results of these comparisons address different aspects within “X- upregulation” in which the X profile is distinct from that of each autosome. We feel both figure panels visualize interesting results that are justified to be shown.

However, concerning Comments 7 and 8, if deemed mandatory we will carry out the proposed changes in figures.